

# Radiolarian assemblages in the shelf area of the East China Sea and Yellow Sea and their ecological indication of the Kuroshio Current derivative branches

Hanxue Qu[1,2,3,4,*], Yong Xu[1,3,4,5,*], Jinbao Wang[1,3,4,5] and Xin-Zheng Li[1,3,4,5]

[1] Institute of Oceanology, Chinese Academy of Sciences, Qingdao, China
[2] Pilot National Laboratory for Marine Science and Technology (Qingdao), Qingdao, China
[3] University of Chinese Academy of Sciences, Beijing, China
[4] Center for Ocean Mega-Science, Chinese Academy of Sciences, Qingdao, China
[5] Laboratory for Marine Biology and Biotechnology, Pilot National Laboratory for Marine Science and Technology (Qingdao), Qingdao, China
[*] These authors contributed equally to this work.

## ABSTRACT

We analyzed the radiolarian assemblages of 59 surface sediment samples collected from the Yellow Sea and East China Sea of the northwestern Pacific. In the study region, the Kuroshio Current and its derivative branches exerted a crucial impact on radiolarian composition and distribution. Radiolarians in the Yellow Sea shelf showed a quite low abundance as no tests were found in 15 of 25 Yellow Sea samples. Radiolarians in the East China Sea shelf could be divided into three regional groups: the East China Sea north region group, the East China Sea middle region group, and the East China Sea south region group. The results of the redundancy analysis suggested that the Sea Surface Temperature and Sea Surface Salinity were primary environmental variables explaining species-environment relationship. The gradients of temperature, salinity, and species diversity reflect the powerful influence of the Kuroshio Current in the study area.

## INTRODUCTION

Polycystine Radiolaria (hereafter Radiolaria), with a high diversity of 1192 Cenozoic fossil to Recent species, are a crucial group of marine planktonic protists (*Lazarus et al., 2015*; *Suzuki, 2016*). Living Radiolaria are widely distributed throughout the shallow-to-open oceans (*Lombari & Boden, 1985*; *Wang, 2012*), and a proportion of their siliceous skeletons settle on the seafloor after death (*Takahashi, 1981*; *Yasudomi et al., 2014*). The distribution of Radiolaria in a given region is associated with the pattern of water mass, such as temperature, salinity and nutrients (*Abelmann & Nimmergut, 2005*; *Anderson, 1983*; *Hernández-Almeida et al., 2017*).

The East China Sea (ECS) and Yellow Sea (YS) are marginal seas of the northwestern Pacific (*Xu et al., 2011*). The two regions are divided by the line connecting the northern

Corresponding author
Xin-Zheng Li, lixzh@qdio.ac.cn

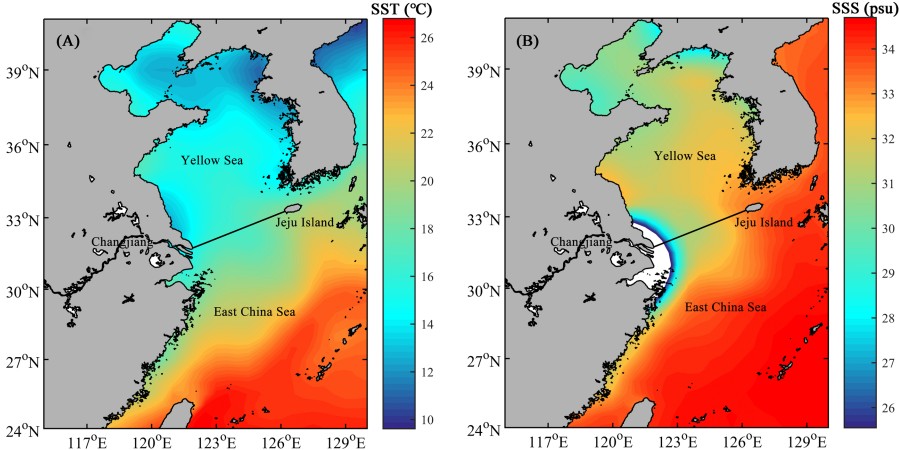

**Figure 1** **The mean annual sea surface temperature (SST, A) and sea surface salinity (SSS, B) in the shelf area of the ECS and YS.** Solid line indicates the boundary between the ECS and YS.

tip of the mouth of the Changjiang and the southern tip of the Jeju Island (*Jun, 2014*). Hydrographic conditions of the shelf area of both the ECS and YS, where the depth is generally less than 100 m, vary remarkably with the season (*Qi, 2014*). Generally, the annual sea surface temperature (SST) and sea surface salinity (SSS) show a decreasing trend from the southeast to northwest in study area (Fig. 1).

The Kuroshio Current originates from the Philippine Sea, flows through the ECS, and afterwards forms the Kuroshio Extension (*Hsueh, 2000*; *Qiu, 2001*). The Kuroshio Current and its derivative branch-the Taiwan Warm Current (TWC), form the main circulation systems in the ECS shelf area, while the Yellow Sea Warm Current, one derivative branch of the Kuroshio Current, dominates in the YS shelf area (*Hsueh, 2000*; *Tomczak & Godfrey, 2001*).

In the ECS shelf region's summer (Fig. 2A), the Kuroshio subsurface water gradually upwells northwestward from east of Taiwan, and finally reaches 30.5°N off the Changjiang estuary along ~60 m isobaths, forming the Nearshore Kuroshio Branch Current (*Yang et al., 2012*; *Yang et al., 2011*). Meanwhile, the TWC is formed by the mixing of the Taiwan Strait Warm Current and Kuroshio Surface Water (*Qi, 2014*). In winter (Fig. 2B), the Kuroshio Surface Water shows relatively intense intrusion as part of the Kuroshio Surface Water northwestward reaches continental shelf area across 100 m isobaths (*Zhao & Liu, 2015*). At this point, the TWC is mainly fed from the Kuroshio Current northeast of Taiwan (*Qi, 2014*).

In the YS shelf region's summer (Fig. 2A), the Yellow Sea Cold Water Mass, characterized by low temperature, occupies the central low-lying area mostly below the 50 m isobaths while the Yellow Sea Warm Current shows little influence (*Guan, 1963*). In winter (Fig. 2B), the impact of the Yellow Sea Warm Current on shelf region is enhanced, while the Yellow Sea Cold Water Mass disappears (*Weng et al., 1988*). The continuous water circulation in the YS is mainly comprised of the Yellow Sea Warm Current and the China Coastal Current (*UNEP , 2005*).

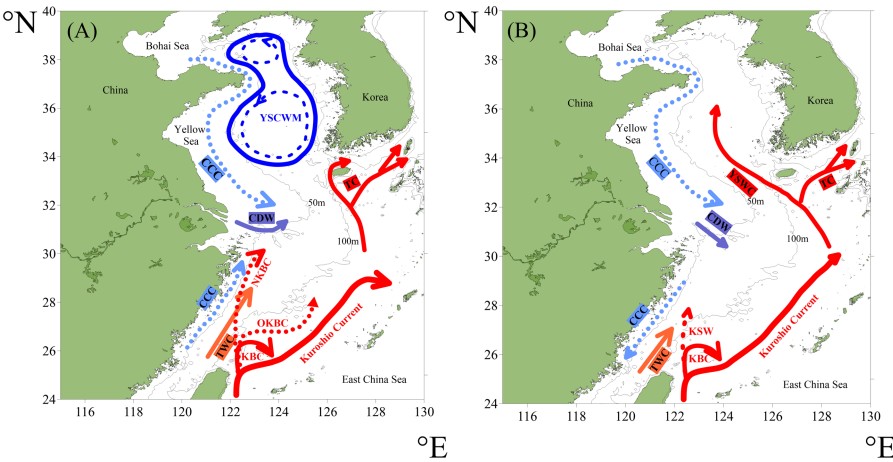

**Figure 2  The circulation system of the study area in summer (A) and winter (B) (redrawn after *Yang et al. (2012)* and *Pi (2016)*).** Abbreviations: KBC—Kuroshio Branch Current, OKBC—Offshore Kuroshio Branch Current, NKBC—Nearshore Kuroshio Branch Current, KSW—Kuroshio Surface Water, TWC—Taiwan Warm Current, CCC—China Coastal Current, CDW—Changjiang Diluted Water, YSCWM—Yellow Sea Cold Water Mass, YSWC—Yellow Sea Warm Current, TC—Tsushima Current.

The radiolarian assemblages in surface sediments have been investigated in the ECS whereas there are few reports in the YS. These reports cover the ECS including the Okinawa Trough (*Chang et al., 2003*; *Cheng & Ju, 1998*; *Wang & Chen, 1996*) and continental shelf region extensively (*Chen & Wang, 1982*; *Tan & Chen, 1999*; *Tan & Su, 1982*). They summarize the distribution patterns of the dominant species and the environmental conditions that affect the composition of radiolarian fauna in the ECS in their excellent taxonomic works. On the basis of these valuable works, we rigorously investigate the relationships between radiolarians and environmental variables. In addition, to which the ECS and YS are influenced by the Kuroshio Current and its derivative branch are specially focused in this study. The radiolarian data collected from 59 surface sediment samples are associated with environmental variables of the upper water to explore the principal variables explaining radiolarian species composition. The influences of the Kuroshio Current and its derivative branch on radiolarian assemblages in the study area are also considerably discussed.

## MATERIALS & METHODS

### Sample collection and treatment

The surface sediments were collected at 59 sites (Fig. 3A) in the Yellow Sea and East China Sea using a box corer. The sediment samples in the study area were divided into four groups geographically and were labeled the Yellow Sea region (YSR) samples, the ECS north region (ECSNR) samples, the ECS middle region (ECSMR) samples, and the ECS south region (ECSSR) samples. The samples were prepared using the method described by *Chen et al. (2008)*. Thirty percent hydrogen peroxide and 10% hydrochloric acid were added to each dry sample to remove organic component and the calcium tests, respectively.

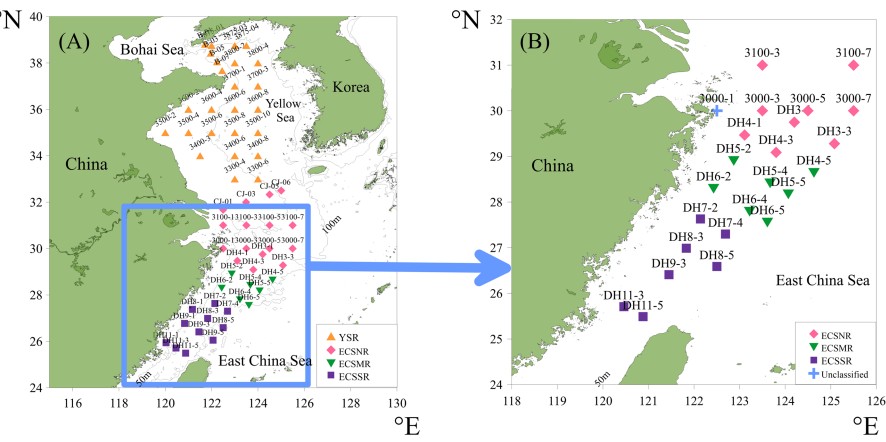

**Figure 3** The location of the total surface sediment samples in the ECS and YS shelf area (A), and simplified 24 samples with a threshold of 100 tests (B).

Then the treated sample was sieved with a 50 μm sieve and dried in an oven. After flotation in carbon tetrachloride, the cleaned residue was sealed with Canada balsam for radiolarian identification and quantification under a light microscope with a magnification of 200X or 400X. To reduce counting uncertainty, *Dictyocoryne profunda* Ehrenberg, *Dictyocoryne truncatum* (Ehrenberg), *Dictyocoryne bandaicum* (Harting) were combined as *Dictyocoryne* group. Photographs of some radiolarians encountered in this study are exhibited in Fig. 4.

## Environmental data

Grain size analysis of the surface sediments was conducted with a Laser Diffraction Particle Size Analyzer (Cilas 1190, CILAS, Orleans, Loiret, France). The data were used to categorise grain size classes as clay (1–4 μm), silt (4–63 μm) and sand (63–500 μm), and to determine different sediment types according to the Folk classification (*Folk, Andrews & Lewis, 1970*). In addition, the mean grain size was calculated for each site.

The values of annual temperature (SST), salinity (SSS), oxygen, phosphate, nitrate, and silicate of sea surface with a 0.5° resolution for the period of 1930 to 2009 were derived from the CARS2009 dataset (*Ridgway, Dunn & Wilkin, 2002*). The sea surface chlorophyll-*a* and particulate organic carbon with a 9 km resolution for the period of 1997 to 2010 were obtained from https://oceancolor.gsfc.nasa.gov/l3/. The values of the environmental variables mentioned above for each surface sediment site were estimated by linear interpolation. These values, together with depth, are shown in Table S1.

## Statistical processing

The minimum number of specimens counted in each sample is customarily 300. However, low radiolarian concentrations are frequent in the shelf type sediments comprised mainly of terrigenous sources (*Chen et al., 2008*). Given small sediment samples, it was difficult to find 300 tests in some sites. According to *Fatela & Taborda (2002)*, counting 100 tests allows less than 5% probability of losing those species with a proportion of 3%. Balanced between the insufficient samples and the accuracy of the statistical analysis, the threshold

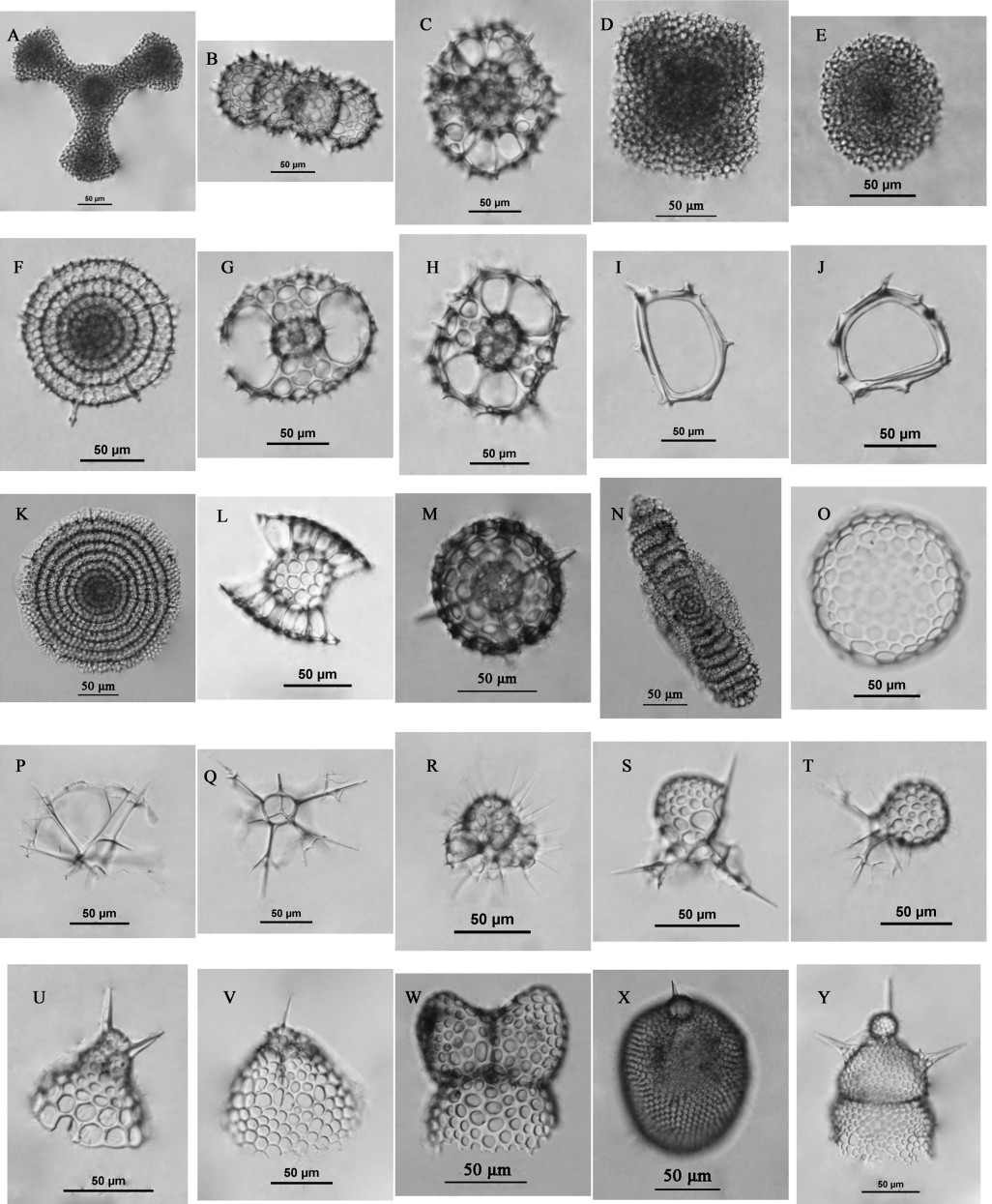

**Figure 4** **Some radiolarians encountered in this study.** (A) *Dictyocoryne* group; (B) *Didymocyrtis tetrathalamus* (Haeckel); (C) *Phorticium pylonium* Haeckel; (D) *Spongaster tetras* Ehrenberg; (E) *Spongodiscus resurgens* Ehrenberg; (F) *Stylodictya multispina* Haeckel; (G–H) *Tetrapyle octacantha* group Müller; (I–J) *Zygocircus piscicaudatus* Popofsky; (K) *Flustrella polygonia* (Popofsky); (L) *Sethodiscus macrococcus* Haeckel; (M) *Hexacontium pachydermum* Jorgensen; (N) *Amphibrachium sponguroides* Haeckel; (O) *Collosphaera* sp.; (P–Q) *Pseudocubus obeliscus* Haeckel; (R) *Acanthocorys castanoides* Tan & Tchang; (S) *Peromelissa spinosissima* Tan & Tchang; T, *Peridium* sp.; (U) *Cycladophora bicornis* (Popofsky); (V) *Helotholus histricosa* Jorgensen; (W) *Phormospyris stabilis stabilis* (Goll); (X) *Lithopera bacca* Ehrenberg; (Y) *Lipmanella dictyoceras* (Haeckel). Scale bar = 50 μm.

number of radiolarians was adjusted to 100 (*Fatela & Taborda, 2002*; *Rogers, 2016*). Based on this threshold, 24 samples (Fig. 3B) were retained for detailed statistical analysis. Seven of 24 samples had less than 300 tests, containing six ECSNR samples and one ECSSR sample. The proportion of each dominant species in the ECSNR group was higher than 3%, guaranteeing a reliable interpretation of species proportions.

We calculated the absolute abundance (tests.$(100 \text{ g})^{-1}$) and the diversity indices, including the species number ($S$), Shannon-Wiener's index ($H'$ ($\log_e$)). To ensure a creditable estimate of diversity indices, which may be biased by different counting numbers, the specimens of radiolarians in each sample were randomly subsampled and normalized to the equal size of 100 tests by using rrarefy() function in vegan package in R program. For each site, $S$ and $H'$ of sample containing all tests and subsample containing 100 tests were calculated.

Relative abundance (%) of each radiolarian taxon was also calculated. Then the hierarchical cluster analysis with group-average linking was applied to analyze the variations of radiolarian assemblage among different regions. The percentage data of the relative abundance was transformed by square root for normalize the dataset. Afterwards, triangular resemblance matrix was constructed based on the Bray-Curtis similarity (*Clarke & Warwick, 2001*). Analysis of similarity (ANOSIM) was employed to determine the differences among different assemblages. Similarity percentage procedure (SIMPER) analysis was used to identify the species that contributed most to the similarities among radiolarian assemblages.

Detrended correspondence analysis (DCA) was applied to determine the character of the species data. The gradient length of the first DCA axis was $1.773 < 3$, suggesting that redundancy analysis (RDA, linear ordination method) was more suitable than canonical correspondence analysis (CCA, unimodal ordination method) (*Lepš & Šmilauer, 2003*). RDA was used to evaluate the relationship between environmental variables and radiolarian assemblages identified by SIMPER analysis. The species abundance data was square root transformed before analysis to reduce the effect of extremely high values (*Ter Braak & Smilauer, 2002*). Variance inflation factor (VIF) was calculated to screen the environmental variables with VIF $> 5$ (*Lomax & Hahs-Vaughn, 2013*). Sand percentage, mean grain size, chlorophyll-*a*, silicate, particulate organic carbon, oxygen, depth, nitrate, and silt percentage were removed from the RDA model step by step, in order to avoid collinearity (*Naimi et al., 2014*). Finally, four variables, SST, SSS, clay percentage, and phosphate, were employed in the RDA. The significant environmental variables were determined by automatic forward selection with Monte Carlo tests (999 permutations). Station DH 8-5 was excluded from the RDA model for lack of environmental data.

Correlation analysis was employed to investigate the relationship between the dominant radiolarian taxa and significant environmental variables.

The diversity indices calculation, cluster analysis, ANOSIM, and SIMPER were performed by PRIMER 6.0. Correlation analysis was performed by SPSS 20. DCA and RDA were conducted by CANOCO 4.5. The detailed description of the statistical methods, including the cluster analysis, DCA, and RDA are in Article S1.

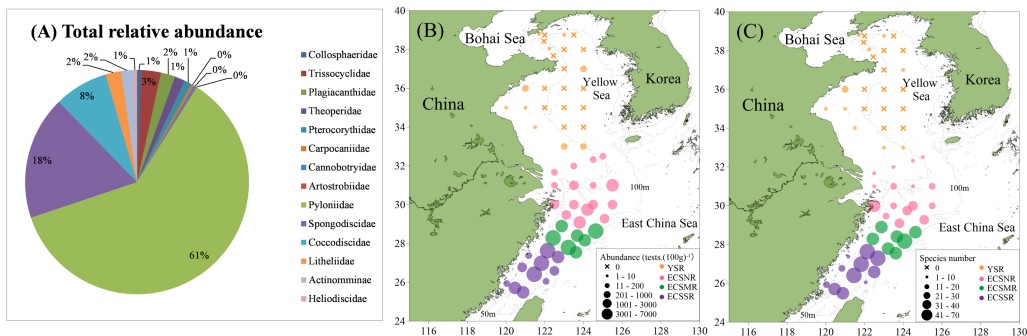

**Figure 5** Total relative abundance (A), absolute abundance (B), and species number (C) of the radiolarians in the surface sediments of the ECS and YS shelf area.

## RESULTS

A total of 137 radiolarian taxa were identified from the surface sediments of study area, including 75 genera, 14 families, and three orders The raw radiolarian counting data are in Table S2. Approximately 91.0% of the species belonged to Spumellaria, accounting for the vast majority of the radiolarian fauna. Nassellaria and Collodaria accounted for 8.4% and 0.6%, respectively. Pyloniidae definitely dominated in the species composition as they occupied approximately 61%; they are followed by Spongodiscidae 18%, and Coccodiscidae 8% (Fig. 5A).

Radiolarian abundance in surface sediments varied greatly in study area (Fig. 5B), showing a tendency of ECSMR (2776 tests. $(100 \text{ g})^{-1}$) > ECSSR (1776 tests. $(100 \text{ g})^{-1}$) > ECSNR (500 tests. $(100 \text{ g})^{-1}$) > YSR (8 tests. $(100 \text{ g})^{-1}$). The distribution pattern of species number (Fig. 5C) was similar to that of the abundance, exhibiting a trend of ECSMR (38 species) > ECSSR (35 species) > ECSNR (16 species) > YSR (1 species). The top nine species taxa, accounting for 79.6% of the total assemblages in the study area, were as follows: *Tetrapyle octacantha* group Müller (55.6%), *Didymocyrtis tetrathalamus* (Haeckel) (7.5%), *Dictyocoryne* group (3.7%), *Spongaster tetras* Ehrenberg (2.5%), *Stylodictya multispina* Haeckel (2.2%), *Spongodiscus resurgens* Ehrenberg (2.2%), *Zygocircus piscicaudatus* Popofsky (2.1%), *Phorticium pylonium* Haeckel (2.0%), and *Euchitonia furcata* Ehrenberg (1.8%).

### The radiolarian assemblages in the YS shelf area

In general, radiolarians showed a quite low abundance value in the YS, as no tests were found in 15 samples (Fig. 5). For the remaining 10 samples, only 49 tests were originally counted, belonging to 21 species taxa. The radiolarian abundance for 25 samples of the YS ranged from 0 tests. $(100 \text{ g})^{-1}$ to 91 tests. $(100 \text{ g})^{-1}$, and species number ranged from 0 to 12. *Tetrapyle octacantha* (17.4%), *Spongodiscus* sp. (10.9%), *Didymocyrtis tetrathalamus* (9.1%), *Acrosphaera spinosa* (6.1%), and *P. pylonium* (6.1%) were the five most abundant species taxa in the YS, constituting 49.7% of the total assemblages.
**Table 1  The average values and standard errors (mean ± SE) of abundance and diversity indices in different regions (ECSNR, ECSMR, ECSSR).**

| Diversity index | ECSNR ($n = 9$) | ECSMR ($n = 7$) | ECSSR ($n = 7$) |
|---|---|---|---|
| $N$ | $811 \pm 121^a$ | $2776 \pm 463^b$ | $2729 \pm 770^c$ |
| $S$ | $21 \pm 1^a$ | $38 \pm 1^b$ | $48 \pm 5^b$ |
| $H'$ | $1.35 \pm 0.10^a$ | $1.61 \pm 0.13^b$ | $2.65 \pm 0.08^c$ |
| $S_{sub}$ | $11 \pm 1^a$ | $16 \pm 1^b$ | $26 \pm 2^c$ |
| $H'_{sub}$ | $1.22 \pm 0.11^a$ | $1.35 \pm 0.13^b$ | $2.43 \pm 0.10^c$ |

**Notes.**

Different lowercase a, b and c indicate significant differences among regional groups.

Abbreviations: $N$, Abundance (tests. $(100\ g)^{-1}$); $S$, species number; $H'$ ($\log_e$), Shannon-Wiener's index; $S_{sub}$, species number of subsamples; $H'_{sub}$ ($\log_e$), Shannon-Wiener's index of subsamples.

## Selected stations in the ECS shelf area with radiolarian tests ≥ 100

As can be seen in Table 1, there exists a significant difference in radiolarian abundance between the three regions (ANOVA, $p = 0.001$). Diversity indices, including $S$ and $H'$, displayed an overall ranking of ECSSR > ECSMR > ECSNR both in samples ($S$, Kruskal-Wallis Test, $p = 0.000$; $H'$, ANOVA, $p = 0.000$) and subsamples ($S_{sub}$, ANOVA, $p = 0.000$; $H'_{sub}$, ANOVA, $p = 0.000$).

Cluster analysis based on the relative abundance classified all but one site into three regional groups at the 60% Bray-Curtis similarity level, including the ECSNR group, ECSMR group and ECSSR group (Fig. 6). The significant differences among the three groups were examined by ANOSIM (Global $R = 0.769$, $p = 0.001$).

The dominant species in each regional group were identified by SIMPER analysis with a cut-off of 50% (Table 2). *Tetrapyle octacantha*, *Didymocyrtis tetrathalamus*, and *Spongodiscus resurgens* dominated in the ECSNR group, with contribution of 41.70%, 9.79%, and 8.89%, respectively. The radiolarian taxa, including *T. octacantha*, *Didymocyrtis tetrathalamus*, *Dictyocoryne* group, *Stylodictya multispina*, and *Spongodiscus resurgens*, contributed most to the ECSMR group. The dominant species in the ECSSR group were composed of *T. octacantha*, *Didymocyrtis tetrathalamus*, *Dictyocoryne* group, *Spongaster tetras*, *Z. piscicaudatus*, *P. pylonium*, *Stylodictya multispina*, and *E. furcata*.

The first two RDA axes explained 39.9% (RDA1 30.0%, RDA2 9.9%) of the species variance, and 86.5% of the species-environment relation variance (Table 3A). Forward selection with Monte Carlo test (999 Permutation) revealed that SST and SSS were the most significant environmental variables associated with radiolarian composition (Table 3B).

The RDA plot showed a clear distribution pattern of regional samples (Fig. 7A). The ECSNR samples generally occupied the upper left-hand quarter of the ordination, showing a feature of comparatively low SST and low SSS. The ECSMR samples were mostly located in the ordination's centre, suggesting an adaptation to higher values of SST and SSS than the ECSNR samples. The ECSSR samples distributed mainly at lower right-hand quarter, characterized by the highest values of SST and SSS.

The dominant species identified by the SIMPER analysis (Table 2) are displayed in the RDA plot (Fig. 7B). Species taxa, including *Spongaster tetras*, *Dictyocoryne* group, *Z. piscicaudatus*, *E. furcata*, *P. pylonium* and *Stylodictya multispina*, were related to high SST,

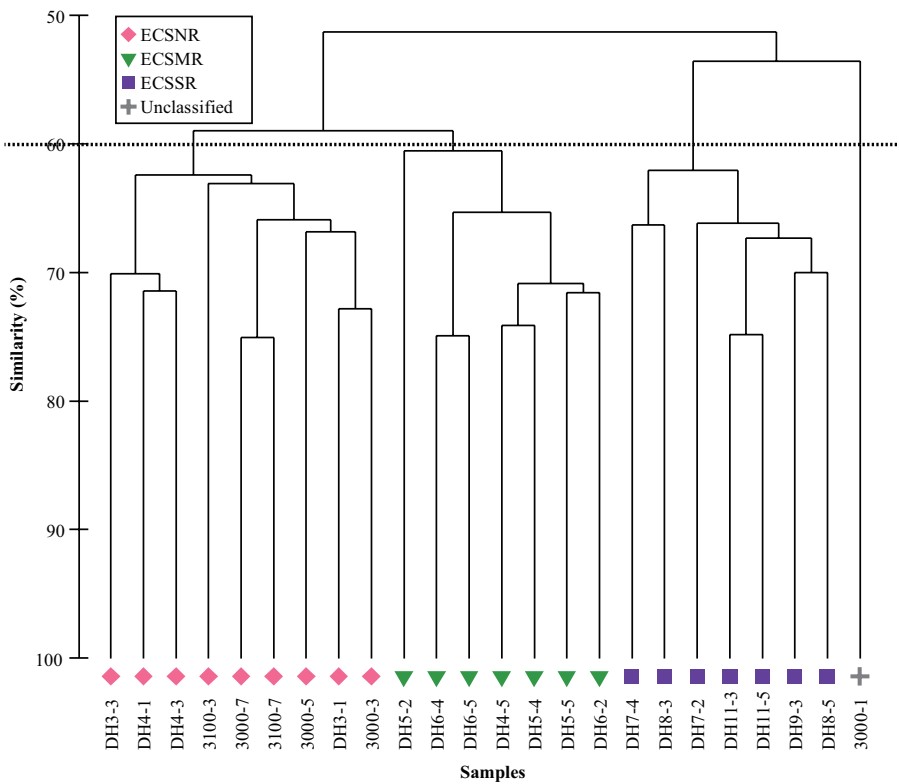

**Figure 6  Cluster analysis of radiolarian assemblages in the ECSNR, ECSMR and ECSSR.** The dotted line represents 60% similarity level.

while showed little relationship with SSS. *Didymocyrtis tetrathalamus* was positively related to SST and SSS. *Tetrapyle octacantha* showed a preference for high SSS. Additionally, *Spongodiscus resurgens* was adapted to relatively low SST and SSS.

## DISCUSSION

Generally, the number of the radiolarian tests in continental shelf sediments of the ECS and YS is several orders of magnitude lower than that of the adjacent Okinawa trough (*Chang et al., 2003*; *Cheng & Ju, 1998*). Firstly, due to the continental runoff input, the coastal area water is characterised by relatively low temperature and salinity, resulting in few living radiolarians (*Chen & Wang, 1982*; *Matsuzaki, Itaki & Kimoto, 2016*; *Tan & Su, 1982*). Also, the deposition rate in study area is high at 0.1–0.8 cm/yr in the YS, and 0.1–3 cm/yr in the ECS (*Dong, 2011*), which greatly masks the concentration of radiolarian skeleton in sediments (*Chang et al., 2003*).

### The radiolarian assemblages in the YS shelf area

Based on our results, although radiolarian assemblages varied greatly between the YS and ECS, there are some common species as all of the 21 radiolarian species in the YS can be found in the ECS, that is, no endemic species were observed in the YS. The five most abundant species taxa, except *Spongodiscus* sp., were reported as typical warm species

**Table 2** Average relative abundance, contribution (%) and cumulative contribution (%) of the radiolarian fauna contributing to the similarity within each group. A cut-off at 50% similarity was employed.

| Species | Av.Abund. | Contrib % | Cum % |
|---|---|---|---|
| ECSNR group | Average similarity: 64.65% | | |
| *Tetrapyle octacantha* group Mueller | 69.67 | 41.70 | 41.70 |
| *Didymocyrtis tetrathalamus* (Haeckel) | 5.65 | 9.79 | 51.49 |
| *Spongodiscus resurgens* Ehrenberg | 4.81 | 8.89 | 60.38 |
| ECSMR group | Average similarity: 66.17% | | |
| *Tetrapyle octacantha* group Mueller | 67.24 | 31.00 | 31.00 |
| *Didymocyrtis tetrathalamus* (Haeckel) | 5.54 | 8.11 | 39.11 |
| *Dictyocoryne* group | 2.53 | 4.62 | 43.73 |
| *Stylodictya multispina* Haeckel | 1.95 | 4.05 | 47.78 |
| *Spongodiscus resurgens* Ehrenberg | 1.55 | 3.96 | 51.74 |
| ECSSR group | Average similarity: 65.02% | | |
| *Tetrapyle octacantha* group Mueller | 38.91 | 17.85 | 17.85 |
| *Didymocyrtis tetrathalamus* (Haeckel) | 9.06 | 7.41 | 25.25 |
| *Dictyocoryne* group | 5.26 | 5.95 | 31.20 |
| *Spongaster tetras* Ehrenberg | 3.98 | 5.01 | 36.21 |
| *Zygocircus piscicaudatus* Popofsky | 3.42 | 4.56 | 40.77 |
| *Phorticium pylonium* Haeckel | 2.98 | 4.43 | 45.20 |
| *Stylodictya multispina* Haeckel | 3.67 | 4.28 | 49.48 |
| *Euchitonia furcata* Ehrenberg | 2.44 | 3.87 | 53.35 |

**Table 3** (A) Results of the RDA for the radiolarian assemblages and environmental variables. (B) Conditional effects of the total environmental variables in the RDA with the significant variables in bold.

| (A) | Axes | | | | Total Inertia |
|---|---|---|---|---|---|
| | **1** | **2** | **3** | **4** | |
| Eigenvalues | 0.300 | 0.099 | 0.039 | 0.024 | 1 |
| Species-environment correlations | 0.955 | 0.971 | 0.816 | 0.772 | |
| Cumulative percentage variance of species data | 30.0 | 39.9 | 43.8 | 46.2 | |
| Cumulative percentage variance of species-environment relation | 65.1 | 86.5 | 94.8 | 100 | |
| Sum of all eigenvalues | | | | | 1 |
| Sum of all canonical eigenvalues | | | | | 0.462 |

| (B) | Conditional effects | | % contribution to canonical eigenvalues | *p* | *F* |
|---|---|---|---|---|---|
| | **VIF** | **LambdaA** | | | |
| **SST** | 1.86 | 0.14 | 30% | **0.004** | 3.34 |
| **SSS** | 2.76 | 0.24 | 52% | **0.001** | 8.01 |
| Clay% | 1.02 | 0.05 | 11% | 0.086 | 1.43 |
| Phospate | 2.69 | 0.03 | 6% | 0.301 | 1.16 |

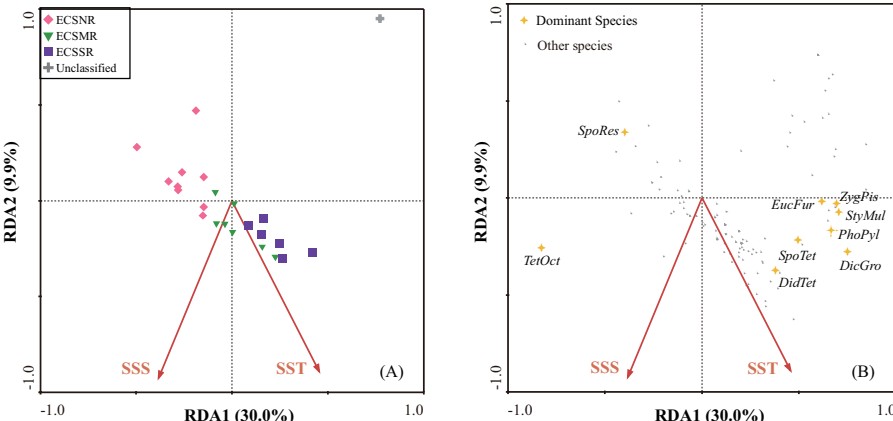

**Figure 7** **The redundancy analysis (RDA) ordination: (A) samples, (B) species.** Species codes: *Dic-Gro*—*Dictyocoryne* group, *DidTet*—*Didymocyrtis tetrathalamus*, *EucFur*—*Euchitonia furcata*, *PhoPyl*—*Phorticium pylonium*, *SpoTet*—*Spongaster tetras*, *SpoRes*—*Spongodiscus resurgens*, *StyMul*—*Stylodictya multispina*, *TetOct*—*Tetrapyle octacantha*, *ZygPis*—*Zygocircus piscicaudatus* (in alphabetical order).

(*Chang et al., 2003*; *Chen et al., 2008*; *Matsuzaki & Itaki, 2017*), suggesting a warm-water origin of radiolarians in the YS.

As a semi-enclosed marginal sea mostly shallower than 80 m, YS is influenced by a continuous circulation, primarily composed of the Yellow Sea Warm Current and China Coastal Current (*UNEP, 2005*). The mean values of SST and SSS in the YS are 15 °C and 32psu, respectively (Fig. 1), making it quite difficult for radiolarians to survive and proliferate. For the surface sediments in the YS in our study, only a small number of radiolarians were detected at the margin of the YS shelf area, whereas no radiolarians were detected in the 15 sites within the range of the central YS (Fig. 5B). For the planktonic samples in the southern YS, low radiolarian stocks were also reported previously (*Tan & Chen, 1999*). Sporadic radiolarians were merely documented in winter, with radiolarian stocks less than 200 tests. $m^{-3}$ (*Tan & Chen, 1999*). We thus infer the radiolarians in the YS (Fig. 5) were probably introduced by the Yellow Sea Warm Current, and transported by the China Coastal Current. Whether the absence of radiolarians in the central YS is controlled by the Yellow Sea Cold Water Mass remains unclear and needs future investigation.

## Selected stations in the ECS shelf area with radiolarian tests ≥ 100

In the ECS, the gradients of SST and SSS are controlled by the interaction of the Kuroshio branch current, TWC and Changjiang Diluted Water (*Yang et al., 2012*). SST and SSS both show an increase from north to south, corresponding well with the overall distribution of radiolarians (Figs. 1 and 5).

As revealed by the RDA, SST was the most significant environmental variable related to the radiolarian composition, followed by SSS (Table 3B). SST is generally regarded as having an extremely important role in controlling the composition and distribution of radiolarians (*Boltovskoy & Correa, 2017*; *Hernández-Almeida et al., 2017*; *Ikenoue et al., 2015*). According to *Matsuzaki, Itaki & Tada (2019)*, the species diversity in the northern

**Table 4** The Spearman correlation between diversity indices and the environmental variables. Values of significant correlations are in bold.

| Diversity index | $n$ | SST | | SSS | |
|---|---|---|---|---|---|
| | | $r$ | $p$ | $r$ | $p$ |
| $N$ | 24 | **0.60** | **0.00** | **0.47** | **0.02** |
| $S$ | 24 | **0.69** | **0.00** | **0.55** | **0.01** |
| $H'$ | 24 | **0.50** | **0.01** | 0.11 | 0.62 |
| $S_{sub}$ | 24 | **0.60** | **0.00** | 0.28 | 0.19 |
| $H'_{sub}$ | 24 | **0.41** | **0.04** | 0.02 | 0.92 |

Notes.

Different lowercase a, b and c indicate significant differences among regional groups.

Abbreviations: $N$, Abundance (tests. $(100 \text{ g})^{-1}$); $S$, species number; $H'$ ($\log_e$), Shannon-Wiener's index; $S_{sub}$, species number of subsamples; $H'_{sub}$ ($\log_e$), Shannon-Wiener's index of subsamples.

ECS was higher during interglacial period than during glacial period. For a long time, the relationship between radiolarian assemblages and SST has been used to construct past changes in hydrographic conditions (*Matsuzaki & Itaki, 2017*). In this study, SST showed a significant correlation with abundance, species number, and $H'$ (Table 4), suggesting that higher SST may often correspond to higher diversity.

SSS was also crucial for explaining species-environment correlations in the ECS shelf area. At the offshore Western Australia, salinity is strongly significant in determining radiolarian species distributions (*Rogers, 2016*). *Hernández-Almeida et al. (2017)* and *Liu et al. (2017a)* stated that the composition and distribution pattern of the radiolarian fauna in the western Pacific responds mainly to SST and SSS. *Gupta (2002)* found that the relative abundance of Pyloniidae exhibits a positive correlation with salinity. In this study SSS was positively correlated to abundance and species number (Table 4), possibly suggesting a positive influence of SSS on radiolarian diversity.

The radiolarian assemblages of the ECSSR group are influenced by the Kuroshio Current and TWC, with the TWC predominating. The surface water of the TWC is mainly characterised by high temperature (23−29 °C) and salinity (33.3–34.2 psu) (*Weng & Wang, 1988*). Some of the TWC waters are supplemented from the South China Sea (*Liu et al., 2017b*), where radiolarians show high diversity (*Chen et al., 2008*; *Liu et al., 2017a*; *Zhang et al., 2009*). The dominant species in the ECSSR group include *T. octacantha*, *Didymocyrtis tetrathalamus*, *Dictyocoryne* group, *Spongaster tetras*, *Z. piscicaudatus*, *P. pylonium*, *Stylodictya multispina*, and *E. furcata* (Table 2, Fig. 8). These species taxa are reported as typical indicators of the Kuroshio Current (*Chang et al., 2003*; *Gallagher et al., 2015*; *Liu et al., 2017a*; *Matsuzaki, Itaki & Kimoto, 2016*). The relatively high abundance of these taxa in the study area reflects the influence of the warm Kuroshio and TWC waters. Moreover, moderate percentage (0.91%) of *Pterocorys campanula* Haeckel was detected in the ECSSR group, in contrast with the ECSMR group (0.14%) and ECSNR group (0.06%). Members of *Pterocorys* are shallow-water dwellers, as reported by *Matsuzaki, Itaki & Sugisaki (2020)*. *Pterocorys campanula* frequently occurs and dominates in the South China Sea, whereas there are no reports of the dominance of *P. campanula* in the sediment samples of the ECS (*Chen & Tan, 1996*; *Chen et al., 2008*; *Hu et al., 2015*; *Liu et al., 2017a*).

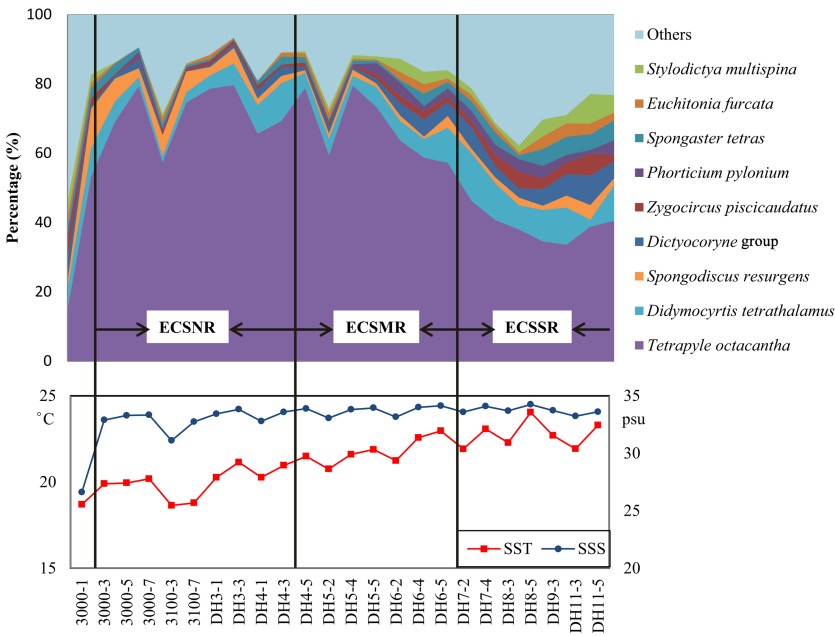

**Figure 8** **Distribution of the dominant radiolarian species, SST, and SSS in the ECSNR, ECSMR, EC-SSR.**

The high abundance of this taxon in the ECSSR group further demonstrates our conclusion that radiolarian assemblages of the ECSSR group are brought by the Kuroshio Current and TWC with the TWC playing the main role.

The ECSMR group was influenced by the Kuroshio Current, TWC, and Changjiang Diluted Water. The dominant species in the ECSMR included *T. octacantha*, *Didymocyrtis tetrathalamus*, *Dictyocoryne* group, *Stylodictya multispina* and *Spongodiscus resurgens* (Table 2). The dominant species of the ECSMR group show a great overlap with the ECSSR group, which, in some degree, suggests a similarity between the two groups, as both are influenced by the Kuroshio Current and TWC. On the other hand, the lower percentages of *Didymocyrtis tetrathalamus*, *Dictyocoryne* group, and *Stylodictya multispina* indicate part of the impact of the Changjiang Diluted Water, which is characterized by lower SST (Fig. 8).

*Tetrapyle octacantha*, *Didymocyrtis tetrathalamus*, and *Spongodiscus resurgens* are the dominant species of the ECSNR group, which is primarily impacted by the Changjiang Diluted Water and Kuroshio Current. Compared to the ECSMR and ECSSR group, the ECSNR group occupied higher latitude which means a lower SST, while the large input of Changjiang Diluted Water lowers the SSS (Fig. 1). This combination of lower SST and SSS probably hindered the radiolarian diversity of the ECSNR (Table 1).

The radiolarian assemblages in the shallower sea, i.e., the shelf sea area of the ECS, displayed distinctly different patterns from those in the open ocean. *Tetrapyle octacantha* occurred in the extraordinarily high proportion of 59% in the study area (Fig. 8), much higher than ever reported in adjacent areas with deeper waters (*Chang et al., 2003*;

*Cheng & Ju, 1998*; *Liu et al., 2017a*; *Wang & Chen, 1996*). The response of *T. octacantha* to SSS was unclear, although it showed positive relationship with SSS in the RDA plot (Fig. 7B). Here a special station with the highest Shannon-Wiener's index (3.2 in both original sample and subsample) was noticed, namely the station 3000-1 (Fig. 3), which is located at the Changjiang estuary. In our study, it had the lowest value of salinity (26.6psu) and the lowest percentage of *T. octacantha* (14.8%). After removing 3000-1, no significant correlation existed between SSS and the relative abundance of *T. octacantha* ($n = 22$, $r = -0.027$, $p = 0.906$). *Tetrapyle octacantha*, as the most abundant taxon in the subtropical area (*Boltovskoy, 1989*), shows a high resistance to SST variation (*Ishitani et al., 2008*). This taxon has been reported to be associated with water from the ECS shelf area (*Chang et al., 2003*; *Itaki, Kimoto & Hasegawa, 2010*). *Welling & Pisias (1998)* concluded that *T. octacantha* dominated during the cold tongue period in the central equatorial Pacific. In the northwest Pacific, there seems to be a threshold value of ~16 °C that only sporadic tests of *T. octacantha* are found with the temperature lower than 16 °C (<6 tests, see Data Set S1 and Table 1 in *Matsuzaki & Itaki, 2017*). In our study, there are very few tests of *T. octacantha* at the temperature below 16 °C (Tables S1 and S2), tending to confirm the former research. We thus infer that *T. octacantha* is possibly more resistant to locally severe temperature and, so, reaches comparatively high abundance in the ECS shelf area. Therefore, *T. octacantha* could serve as an indicator that depicts the degree of mixture between the colder shelf water and warm Kuroshio water. *Spongodiscus resurgens*, with an upper sub-surface maximum, was generally considered to be cold water species (*Suzuki & Not, 2015*) and related to productive nutrient-rich water (*Itaki, Minoshima & Kawahata, 2009*; *Matsuzaki & Itaki, 2017*). The ECSNR group was primarily controlled by the colder Changjiang Diluted Water, and thus had the highest percentage of *T. octacantha* and *Spongodiscus resurgens* among three regions.

## CONCLUSIONS

We analyzed radiolarian assemblages collected from the YS and ECS shelf area, where the Kuroshio Current and its derivative branches, including the TWC and Yellow Sea Warm Current, exerts great effect.

(1) The radiolarian abundance in the YS was quite low, and no radiolarians were detected in 15 of 25 YS sites.

(2) The radiolarian abundance and diversity in the ECS, which is controlled by the Kuroshio warm water, was much higher. Based on the cluster analysis, the radiolarian assemblages in the ECS could be divided into three regional groups, namely the ECSNR group, the ECSMR group and the ECSSR group.

a. The ECSNR group was chiefly impacted by the Changjiang Diluted Water and Kuroshio Current, with dominant species of *T. octacantha*, *Didymocyrtis tetrathalamus*, and *Spongodiscus resurgens*.

b. The ECSMR group was controlled by the Kuroshio Current, TWC and Changjiang Diluted Water. Species contributed most to this group included *T. octacantha*, *Didymocyrtis tetrathalamus*, *Dictyocoryne* group, *Stylodictya multispina*, and *Spongodiscus resurgens*.

c. The ECSSR group was affected by the Kuroshio Current and TWC, in which the TWC occupies major status. The dominant species in this group were composed of *T. octacantha*, *Didymocyrtis tetrathalamus*, *Dictyocoryne* group, *Spongaster tetras*, *Z. piscicaudatus*, *P. pylonium*, *Stylodictya multispina*, and *Euchitonia furcata*.

(3) The RDA results indicated that SST and SSS were main environmental variables that influenced the radiolarian composition in the ECS shelf.

## ACKNOWLEDGEMENTS

We are grateful to laboratory members for collecting sediment samples. We thank Dr. Lanlan Zhang for advising on this research. We also appreciate Dr. Kenji M. Matsuzaki, Dr. John Rogers, and the editor for their valuable comments on the manuscript.

### Funding

This study was supported by the Strategic Priority Research Programs of the Chinese Academy of Sciences (Nos. XDA23050304, XDA11020303), and the Scientific and Technological Innovation Project Financially Supported by Qingdao National Laboratory for Marine Science and Technology (No. 2015ASKJ01). The funders had no role in study design, data collection and analysis, decision to publish, or preparation of the manuscript.

### Grant Disclosures

The following grant information was disclosed by the authors:
Strategic Priority Research Programs of the Chinese Academy of Sciences: XDA23050304, XDA11020303.
Scientific and Technological Innovation Project Financially Supported by Qingdao National Laboratory for Marine Science and Technology: 2015ASKJ01.

### Competing Interests

The authors declare there are no competing interests.

### Author Contributions

- Hanxue Qu conceived and designed the experiments, performed the experiments, analyzed the data, prepared figures and/or tables, authored or reviewed drafts of the paper, and approved the final draft.
- Yong Xu performed the experiments, analyzed the data, authored or reviewed drafts of the paper, and approved the final draft.
- Jinbao Wang performed the experiments, authored or reviewed drafts of the paper, and approved the final draft.
- Xin-Zheng Li conceived and designed the experiments, performed the experiments, authored or reviewed drafts of the paper, and approved the final draft.

### Data Availability

The raw data are available in the Supplementary Materials.

**Supplemental Information**

Supplemental information for this article can be found online at http://dx.doi.org/10.7717/peerj.9976#supplemental-information.

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
