# Peer review of "Radiolarian assemblages in the shelf area of the East China Sea and Yellow Sea and their ecological indication of the Kuroshio Current derivative branches"

_PeerJ, doi:10.7717/peerj.9976_

## Round 0.1 · original submission · Major Revisions

· Academic Editor

Major Revisions

I have heard back from two reviewers, both of whom offer numerous constructive comments on your work. Although not nearly the expert on radiolarians that the reviewers are, I agree with the large majority of their comments and suggestions. I suggest you look these over carefully along with the attachments, and thoroughly rework your manuscript. I look forward to seeing a revised version of your manuscript.

·

Basic reporting

English:
Not too bad but difficult to follow in parts. I have made many comments on the attached pdf.
The paper is probably rather too short but more detailed explanation will help.

References: Sufficient

Article Layout: Poor
o Subheadings not easily distinguishable from the body of the text.
o Lack of new paragraphs.
o Some unnecessary repetition.
o Probabilities quoted as “p < “ do not meet PeerJ standards.

Figures:
Inadequate and sometimes illegible labeling in figures 1-3; otherwise ok.

Tables:
Fine except that amalgamating Supplementary Tables 2 and 3 would make it easier to understand the discussion in the text.

Overall: Self-contained

Other:
Many examples of mixed fonts, mainly (solely?) in species names. This is almost certainly due to copying from the radiolaria.org website.
o Genera & species names should be in italics.
o Lipmanella dictyoceras (Haeckel, 1861) occurs twice in both Supplementary Tables 2 & 3. In Table 3 the census results for the two entries for this species differ - should one of be a different species?

Experimental design

Experimental design

Journal Fit:
Within PeerJ Aims & Scope

Research Question:
Adequate statement. The research is a sensible addition to the body of oceanographic knowledge but is local, rather than global, in its scope.

Investigation:
Fairly standard. Some degree of “overkill” in the number of diversity indices calculated – one or two would have been plenty.

Methods:
Description definitely inadequate. I ran the cluster analysis, the detrended CA, & the RDA but failed to reproduce the authors' results (although my answers were in the same ballpark). Consider that, in addition to their raw data the authors should include a Supplementary Table containing exactly the data they tested & should provide all the parameters they used. The authors used commercial software so should be able to provide all the input parameters.

Validity of the findings

Data:
Not provided in a form which would allow replication.

Conclusions:
Many examples of assuming causation from correlation or similar statistical evidence (e.g. line 233 “Silt percentage significantly affected the radiolarian species composition in study area.”; also lines 223 & 231).

·

Basic reporting

The author provide a good report about radiolarian assemblages changes in the Yellow Sea and the East China Sea and they tried to explain the difference in radiolarian assemblages by a different ecology of the local water masses (Salinity, Temperature...). Thus, it an interesting report.

Experimental design

The design is good and follow the standard methodology and the authors did a considerable effort doing statistical analysis. This is also good.

Validity of the findings

The finding are good and the method also. However, I have seen some discrepancy in the interpretation of Tetrapyle inside their own manuscript and there is also some statistic concerns.

I wonder if the authors checked the multiple collinearity between the environmental variables prior other statistical analysis. Thus, I suggest to revise this issue as it can highly affect the discussion.

The relationship between Tetrapyle and Salinity should be revised as two opposite interpretation is proposed in the manuscript and may lead some confusions.

There is one important taxonomic error, which may affect the discussion. S. resurgens showed by the author is S. biconcavus. This is a concern because S. resurgens inhabit temperate water while S. biconcavus warm water. In addition in the present manuscript, this species group is a dominant species thus, the interpretation may have to change. I suggest to carefully check the spongodiscus data (re-count just this group) and revise it.

Therefore, I think that major revision is most suitable for this time because there are some important things to clarify about the Tetrapyle ecology, and the authors should clarify if Salinity and Temperature are collinear or not.

Additional comments

MS number: 44305
Authors: Hanxue Qu et al.,

The authors discuss changes in radiolarian assemblages in the marginal seas of the Northwest Pacific focussing on the Yellow Sea and East China Sea (Southern and Northern part). Their results are interesting and are helpful for the community of radiolarian paleontologist. However, there are several concerns in the data interpretation, which must be fixed before publication. Thus I recommend major revisions.

Major concerns:

1/ The authors investigated changes in radiolarian assemblages in the East China Sea and in the Yellow Sea, with many samples collected from the Continental Shelf. This area has a shallow water depths and thus the preservation of radiolarian is not good. Thus, as they showed in their supplements, there are a lot of station where radiolarian assemblages is composed of less than 100 specimens. They have neglected those samples from their study, which is good. However, I do not think that it is meaningful to estimate the diversity index in such area because you automatically, you will have a higher diversity in samples where you have counted 300 than in those you counted 100. Thus the proposed index is highly biased. It may be good to provide some assumption and show that you are aware from this issue.

2/ The authors did an effort carrying statistical analysis, which is good. However, it was ambiguous in the text and I did not understand if the authors checked the multiple collinearity between the environmental variables they used (SSS, SST, Grain size, chlorophyll…). This stage is crucial as the authors want to explain changes in radiolarian assemblages based on the 6 environmental variable they retained. I recommend to check first if there is any multiple collinearity in the retain variable by providing a table where the VIF is shown for each. I particularly want to see if the SST and SSS have a multiple collinearity or not. If they are collinear thus the proposed discussion should be modified and say that radiolarians seems to be control by a SSS-SST package hard to be dissociated. If they are really independent variable thus the discussion in it current form is fine.

3/ Interpretation of Tetrapyle spp.: The authors say that Tetrapyle is related to high SSS in the results, however in the discussion they say the Tetrapyle is dominant in an area where you have a mix of fresh water from the Yangtze Rivers discharges with the Kuroshio Current. This is strange as fresh water will diminish the SSS and thus if Tetrapyle is a high SSS marker it should decrease. I think that a careful check in the collinearity between SSS and SST must be done and also a more comprehensive review must be done about Tetrapyle spatio-temporal distribution. Indeed, they also claim that Tetrapyle is a marker of cold water tongue. This is also a strange statement as all the papers published until now shoed that Tetrapyle % increase during interglacial period during the Quaternary in the Japan Sea and in the East China Sea. Please check the paper below:

Matsuzaki, K.M., Itaki, T. and Tada, R., 2019., Paleoceanographic changes in the Northern East China Sea during the last 400 kyr as inferred from radiolarian assemblages (IODP Site U1429). Progress in Earth and Planetary Science, v. 6., no.22, pp. 1-21

Itaki, T., Sagawa, T., and Kubota, Y., 2018. Data report: Pleistocene radiolarian biostratigraphy, IODP Expedition 346 Site U1427. In Tada, R., Murray, R.W., Alvarez Zarikian, C.A., and the Expedition 346 Scientists, Proceedings of the Integrated Ocean Drilling Program, 346: College Station, TX (Integrated Ocean Drilling Program). doi:10.2204/iodp.proc.346.202.2018

Thus re-thinking is probably needed.

4/ There is a major taxonomic error about S. resurgens. The photo showed here indicate that what the authors called S. resurgens is S. biconcavus. This is a concern because S. resurgens is more likely a temperate to cold water species while S. biconcavus inhabited warm area. I recommend to check carefully the counts about spongodiscus species and re-think the interpretation.

Minor concerns:

L.29-31: Should be re-phrased. I do not understand the meaning.

L. 35: I suggest to check the paper of Suzuki (2016,The statistic information on radiolarian studies based on PaleoTax for Windows, a synonym database: Fossil, 99,15‒31). There is more than 1000 species in modern ocean.

L. 37: I also suggest to have a look on Lombard and Boden 1985 Atlas. Reference: Lombari, G., G. Boden (1985), Modern radiolarian global distributions. Cushman Foundation for Foraminiferal Research, Special Publication 16A, 68-69.

L. 38: This is not a representative paper for radiolarian dissolution. I suggest to check:

Takahashi K, (1981) Vertical flux, ecology and dissolution of Radiolaria in tropical oceans: implications for the silica cycle (Doctoral dissertation, Massachusetts Institute of Technology and Woods Hole Oceanographic Institution). DOI:10.1575/1912/2420, https://hdl.handle.net/1912/2420

L. 40: There is more paper about that showing good schemes, see also :

Abelmann, A., Nimmergut, A. (2005). Radiolarians in the Sea of Okhotsk and their ecological implication for paleoenvironmental reconstructions. Deep Sea Research Part II: Topical Studies in Oceanography, 52(16), 2302-2331.

L. 49-51: I do not understood well. The KC do not enter the Yellow Sea but the coastal warm current which bifurcate from Taiwan Warm Current is. So perhaps also look at the book below where the oceanography is described easily for a general scientific audience:

Tomczak, M., Godfrey, J.S. (1994). Regional Oceanography: an Introduction. Pergamon Press, Oxford, 1-422.

L.68-72: What you say is the true, but Chen & Wang, 1982; Tan & Su 1982 are both excellent taxonomically so it may be good to say that they are amazing taxonomic works.

L.73-77: I recommend to focus on clarifying changes in radiolarian assemblages among both sea, which can be explain by changes SST, SSS… It is better to be more focussed on your assemblage data. Otherwise we get lost in your objectives.

L. 88: Canada gum-> Canada balsam

L. 100-1001: Is that a 1° interpolation?

L. 106-114: See my major concern 1 above.

L. 118: it is better to add "for normalize the dataset » after square root.

L. 128-129: I think it is related to my major comment 2. The VIF mean that you check the multiple collinearity between the environmental variable? If so please show all the results in the main text and explain why you took the threshold of VIF>10 usually it is 5 based on Lomax, R. G., & Hahs-Vaughn, D. L. (2012). An introduction to statistical concepts (3rd ed.). Routledge/Taylor & Francis Group.

128-129: Please show data if you want the neglect it.

L. 156: I understand the meaning but please re-phrase it with word.

L. 161-163: Again I am not sure how much is the diversity index suitable in this context.

L. 171-172 and elsewhere: Please when you cite a species for the first time in a manuscript provide the full taxonomic name, which include Genus, species name and author(s).

L. 196: see major comments 3.

L. 202-204: Similar thing has been said by Matsuzaki et al. (2016, Mar Micro) also please check it.

L. 211: How is the water depths influencing it? It may be good to also consider it.

L. 218: Silt percentage mean grain size? if so it may be better to use the term grain size.

L. 222-224: Matsuzaki et al. (2019) also showed that higher diversity is reached during interglacial period in the East China Sea over the last 400 000 years. It is supporting your idea.

L. 230-232: See major comment 2.

L. 236-239: What it is written here is of interest however if the grain size affect the diversity as it is written here, it means that it is likely that the dissolution is affecting the assemblages, with coarser grain sizes causing higher dissolution, which may lead the complete disparition of radiolarians having a weaker Si02 skeletons. Thus I also suggest to consider dissolution effect.

L. 255-256: I suggest to check Matsuzaki et al., in press. for Pterocorys % as they published new living records in the Kuroshio Area (Kyushu Pale-Ridge):

Matsuzaki, K. M., Itaki, T. and Sugisaki, S., in press: Polycystine radiolarians vertical distribution in the subtropical Northwest Pacific during Spring 2015 (KS15-4). Paleontological Research. 10.2517/2019PR019

L. 269-273: See major comment 3.

L. 281-287: See major comment 3.

Taxonomy:

There are some important error in particular S. resurgens-> S. biconcavus. Both species clearly lives in different habitat so I think that a careful check should be done on counting Spongodiscus.

A: This is not a typical D. profunda but more Dictyocoryne bandaicum (Harting) See Matsuzaki et al. (2015 Marine Micropaleontology).
E: This is definitely not S. resurgens. This is Spongodiscus biconcavus Haeckel. The very opaque and thick center with the presence of a pylome at the periphery clearly indicate it. See Matsuzaki et al. (2016).
G-H: Good, but I will rather say T. octacantha Muller group as there is a high variability.
K: This is wrong. This is more Flustrella polygonia (Popofsky)
See: Matsuzaki, K. M., Suzuki, N., and Nishi, H., 2015, Middle to Upper Pleistocene Polycystine Radiolarians from Hole 902-C9001C, Northwestern Pacific: Paleontological Research, v. 19, no. s1, p. 1-77.
O: This is wrong. The pores are to big and the shell to rounded for be a C.Huxleyi. I rather recommend to not name this species ore use sp.
P-Q: This is Pseudocubus obeliscus Haeckel.

---

## Round 0.2 · Minor Revisions

· Academic Editor

Minor Revisions

I have heard back again from both reviewers, who find your work greatly improved. Both have added some small edits, mainly to improve the English, and I anticipate the needed revisions will be easy to complete. I look forward to seeing your revised paper.

·

Basic reporting

English: Much improved on the original version & nothing really difficult to follow. I have made some comments in the attached annotated PDF. I note the enhanced explanations.

References: Appropriate & sufficient.

Article Layout: Much improved & now suitable for publication.

Tables & Figures: Much improved & now suitable for publication.

Overall: Self-contained.

Experimental design

Journal Fit: Within PeerJ Aims & Scope.

Investigation: Good

Methods: The inclusion of the authors' raw data has enabled me to use open-access software to confirm their RDA results as well as their cluster analysis & detrended CA.

Validity of the findings

Data:
There is now enough raw data to allow replication so I have been able to confirm the authors' results.

Conclusions:
The authors' previous wording which frequently suggested correlation meant causation has been amended.

Additional comments

This version of the paper is a great improvement on the first. The only criticism I have is that I still feel rather too many statistical tests have been applied. For example, I am not sure that SIMPER provides any more information than could have been deduced by visual examination of the census data.

Note: Line 31: Key words omitted – were lines 32-33 in previous version.

·

Basic reporting

The manuscript is clear and well-written. The cited literature is sufficient, providing us a good background about the research context. The hypothesis and the data presented here are consistent. To summarize the present manuscript is in a good shape.

Experimental design

The manuscript showed new original data and the method follow the international standards and is well explained. The research is also relevant and have a clear purpose well defined. There are no issues in the design of the research.

Validity of the findings

The finding presented here are relevant and allow the micropaleontological community to have a better understanding of a particular siliceous microfossil group in the East China Sea and the Yellow Sea, which can help to reconstruct hydrography of the past. The used robust statistics and thus it is looking good.

Additional comments

The data is of interest and allow us to have a better understanding of key parameters constraining the distribution and diversity of Radiolarians. The authors also stressed well the importance of not only the Kuroshio Current but also the importance of the Yangtze river discharges of fresh water in the regional ecosystem. Therefore, the authors did an excellent job and the manuscript is in a good shape and I think it is ready for publications.

I have few very minor comments/advices, which may be considered before sending the proofs. They are listed below:

L. 166: Mueller-> Müller
L. 203: adaption-> adaptation
L. 209: According to figure 7, I am not sure that Z. piscicaudatus, E. furcata… are related to lower SSS. They are related to High SST. There is a bit of over interpretation. I will suggest to delete « lower SSS » .
L. 211: Same than above. I think that there is a bit of over interpretation in saying that T. octacantha fit with lower SST. However higher SSS yes. So I suggest to delete « lower SST » .
L. 241-243: What you wrote is good, but because it seems that there is a latitudinal changes in SST and SSS. So how about the possible effect of solar insolation? You may not need to address it in this MS, but may be good for you to keep it in mind for further studies.
L.276: Matsuzaki, Itaki, Sugisaki (2019)->Matsuzaki, Itaki, Sugisaki (2020)
L. 300-301: I agree what you say, it is true if you just consider the tropical marginal seas. However, if you looked at the entire N. Pacific, it seems that there is a threshold value of about 15℃. This means that Tetrapyle spp. is as you say highly resistant to SST variation, but cannot survive to SST lower than about 15 ℃. (Matsuzaki and Itaki, 2017)

---

## Round 0.3 · accepted · Accept

· Academic Editor

Accept

The manuscript has been well revised, and I am very happy to move this into production. Congratulations!